# Influence of the Drying Process on the Volatile Profile of Different *Capsicum* Species

**DOI:** 10.3390/plants13081131

**Published:** 2024-04-18

**Authors:** Cosimo Taiti, Diego Comparini, Lavinia Moscovini, Simona Violino, Corrado Costa, Stefano Mancuso

**Affiliations:** 1Department of Agrifood Production and Environmental Sciences, University of Florence, Viale delle Idee 30, Sesto F.no, 50019 Florence, Italy; cosimo.taiti@unifi.it (C.T.); stefano.mancuso@unifi.it (S.M.); 2Consiglio Per La Ricerca in Agricoltura e L’analisi Dell’economia Agraria (CREA), Centro di Ricerca Ingegneria e Trasformazioni Agroalimentari, Via della Pascolare 16, 00015 Monterotondo (RM), Italy; lavinia.moscovini@crea.gov.it (L.M.); simona.violino@crea.gov.it (S.V.); corrado.costa@crea.gov.it (C.C.); 3Fondazione per il Futuro delle Città, Via Boccaccio 50, 50133 Firenze, Italy

**Keywords:** *Capsicum*, DRYING process, PTR-TOF-MS, PLSDA, volatile analysis

## Abstract

Chili is a globally significant spice used fresh or dried for culinary, condiment, and medicinal purposes. Growing concerns about food safety have increased the demand for high-quality products and non-invasive tools for quality control like origin tracing and safety assurance. Volatile analysis offers a rapid, comprehensive, and safe method for characterizing various food products. Thus, this study aims to assess the impact of the drying process on the aromatic composition of various *Capsicum* species and to identify key compounds driving the aromatic complexity of each genetic makeup. To accomplish these objectives, the aroma was examined in fruits collected from 19 different pepper accessions (*Capsicum* sp.) belonging to four species: one ancestral (*C. chacoense*) and three domesticated pepper species (*C. annuum*, *C. baccatum* and *C. chinense*). Fresh and dried samples were analyzed using a headspace PTR-TOF-MS platform. Our findings reveal significant changes in the composition and concentration of volatile organic compounds (VOCs) from fresh to dried *Capsicum*. Notably, chili peppers of the species *C. chinense* consistently exhibited higher emission intensity and a more complex aroma compared to other species (both fresh and dried). Overall, the data clearly demonstrate that the drying process generally leads to a reduction in the intensity and complexity of the aromatic compounds emitted. Specifically, fresh peppers showed higher volatile organic compounds content compared to dried ones, except for the two sweet peppers studied, which exhibited the opposite behavior. Our analysis underscores the variability in the effect of drying on volatile compound composition among different pepper species and even among different cultivars, highlighting key compounds that could facilitate species classification in dried powder. This research serves as a preliminary guide for promoting the utilization of various pepper species and cultivars as powder, enhancing product valorization.

## 1. Introduction

The genus *Capsicum* belongs to the Solanaceae family, which includes more than 30 species and is grown mostly in tropical and subtropical regions of the world. Since they have many beneficial effects for human health, the production and consumption of chili peppers (both fresh and dried) is very high worldwide. Indeed, chili pepper represents an important nutraceutical food as many beneficial compounds are found within the fruit, such as capsaicinoids, carotenoids, phenolic compounds, flavonoids, vitamins, minerals, essential oils, and aromatic substances. Nowadays, the dried chili pepper (*Capsicum* spp.) is used as a condiment worldwide and falls among the healthiest spices, with an average recommended consumption of 0.5 g/person per day [1]. The drying process aims to increase packaging efficiency, storage, and transportation, making the spice suitable for storage at room temperature [2]. In addition, improving the storage time promotes macronutrient concentration, eliminating the need of additives and preventing fungal proliferation. Moreover, through the drying process, the original organoleptic properties are altered, giving rise to new compounds and sometimes improving the sensorial aspect and quality of other foods [3]. The quality of dried *Capsicum* is evaluated through parameters such as color, spiciness, volatile aroma, and flavor compounds; all these quality parameters play a key role in consumer choice and acceptance [4]. Pungencies and aromas that are linked to the volatile emission are key features in evaluating the consumer acceptance of fruits [5]. Moreover, the volatile organic compounds (VOC) are commonly correlated with flavor and aroma in foodstuffs and are therefore linked with the origin and quality of chili [5,6]. Therefore, as reported by Ko and co-author [7], VOCs determination is very important in assessing food quality and preventing food fraud.

Many studies have reported and identified more than 125 volatile compounds in fresh and processed *Capsicum* fruits; however, the significance of these compounds for the aroma is not yet well understood [8,9]. As reported elsewhere [4,5,10], the VOC compositions and concentrations vary significantly among *Capsicum* species and between fresh and dried samples. In particular, some volatile compounds (i.e., 1-penten-3-one and other C5 or C6 compounds) available in fresh chili can be modified when exposed to thermal and oxidative degradation [11]. The drying process represents a versatile and widespread technique and is the object of continuous interest among food researchers, therefore this paper presents the first study aimed at understanding the volatile profile changes between fresh and dry samples in the three most important domesticated *Capsicum* species and one wild species. To date, it is interesting to note that only a few studies—based on *C. annuum* and mostly on sweet peppers—have been conducted with the aim of identifying how the volatile profile of *Capsicum* fruit changes from fresh to dried states [5,10,12,13,14,15,16]. Indeed, this study represents the first attempt to investigate how the aroma changes from fresh to dry in different *Capsicum* species.

Therefore, this study aims to (1) examine the effects of the drying process on the aroma composition of different species of chili fruits and (2) identify volatile markers for discriminating dried chili species.

## 2. Materials and Methods

### 2.1. Plant Materials

This study was performed on 19 different *Capsicum* accessions belonging to four different species, as reported in Figure 1. The fruits of all varieties studied had been previously described (fruit color, fruit shape, fruit weight, fruit volume, and capsaicin content) by [4]. In particular, cultivars belonging to one ancestral (*C. chacoense*) and three domesticated species were selected (*C. annuum*, *C. baccatum*, and *C. chinense*). Among the domesticated species, two cultivars with an absent or very low capsaicin level, namely, “Sweet chocolate” and “Bell pepper”, were selected (Figure 1). Plants from seeds were grown in plastic pots filled with peat and compost inside a greenhouse with controlled temperatures (25 °C/16 °C day/night). Drip irrigation, fertigation, and pest management were applied as reported by Taiti et al. [4]. For each of the pepper accessions (*n* = 19), six to nine plants were cultivated inside a greenhouse located in the Department of Agrifood Production and Environmental Sciences (University of Florence, Italy). The number of fruits harvested varied depending on fruit size, while the harvesting time varied according to the ripening stage of each fruit. For each pepper cultivar, a total of ~250 g of fully ripe fruits were harvested from 5/10 different plants. The harvested fruits were divided into two lots: 100 g of fresh pepper fruits were used immediately for VOCs analysis (first lot), while the other 150 g were subjected to the drying process (second lot).

### 2.2. Drying Peppers

The dehydration process is an essential step for spice preservation as it is able to reduce the microbial spoilage that causes significant loss of color and shelf life [17]. To reduce the negative effects of dehydration on antioxidant activity, ascorbic acid, and color, a short-time and low-temperature approach is employed during the dehydration process [18]. Therefore, 150 g of fresh pepper fruits were washed, cut into 3 parts, distributed uniformly as a thin layer onto the stainless steel trays, and subsequently dried in a thermostatic oven (M710, Galli, Milan, Italy) for 6 h at 70 °C, following the protocol proposed by Kim and co-authors [18]. At the end of the dehydration process, the moisture content obtained was between 10 and 13% to prevent potential aflatoxin production [19]. Subsequently each sample was pulverized throughout a food processor to obtain powder. The dried powder was packed in a common glass jar and stored individually at room temperature in the dark before being subjected to analysis within two weeks of collection.

### 2.3. Volatiles Organic Compounds (VOCs) from Fresh and Dried Samples

Headspace analysis was performed first on fresh samples and then on dried ones. For each pepper cultivar, the sample was constituted of 10 g of fresh or dried pepper. Each sample was analyzed three times, and the average signal value of each peak was taken and reported as parts per billion by volume (ppbv). A total of 8 fresh samples and 4 dried peppers samples were analyzed for each cultivar studied. At the start and after every three samples, an empty bottle was monitored to obtain the blank value that served as control. The headspace analysis has been performed using a PTR-TOF 8000 model (Ionicon Analytik GmbH, Innsbruck, Austria). The PTR is a powerful analytical technique used for the real-time detection and quantification of volatile organic compounds (VOCs) in various samples. It is a solvent-free tool that efficiently collects data on a large number of volatile compounds with both high and low molecular weights without altering samples. However, like any analytical method, it has its limitations, particularly regarding its capability to detect specific VOCs because its technology relies on the proton transfer reaction between VOCs and hydronium ions (H_3_O^+^). However, not all VOCs have sufficient proton affinity to undergo efficient ionization; some compounds may not ionize or may ionize with low efficiency, resulting in reduced detection sensitivity. A comprehensive description of the instrument is reported elsewhere [20,21]. All samples were analyzed using H_3_O^+^ as a reagent ion, following the same procedure and within a mass spectrum ranging from *m*/*z* 20 to 220. In brief, once the sample was placed inside an airtight jar, volatile analysis was performed for a total of 120 s for each sample, with a flow rate of 50 sccm. To avoid possible contamination among samples, the inlet was continuously replaced with clean air (Zero Air Generator, Peak Scientific, Glasgow, UK) between one measurement and the next. Given the aromatic complexity of some samples, and despite the high mass resolution provided by the tool, in few cases, saturated peaks, as well as overlapping peaks, were found within the mass spectrum considered. In these cases, since the overlap of the ion peak by an intense neighbor could affect the accuracy of that ion measurement, we focused our attention on the isotope peaks. Finally, the number of peaks detected was reduced by applying noise thresholds. In particular, the following were removed: (1) peaks not significantly different from blank samples; (2) signals linked to isotope mass peaks; (3) peaks linked to water chemistry (hydronium ion, water clusters) and other interfering ions (e.g., oxygen, nitrogen monoxide); (4) signals related to ions present in trace amounts with a value under 0.5 ppbv.

### 2.4. Statistic Analysis

Linear and non-linear approaches have been applied in order to classify species (within fresh and dried samples) and fresh or dry matrices (within species).

A partial least squares discriminant analysis (PLSDA) linear approach was used to characterize species or fresh/dried peppers according to the VOCs identified with the PTR-ToF-MS technique. PLSDA consists of a PLS regression analysis in which the response variable is categorical (Y-block; replaced by a set of dummy variables describing the species or the fresh/dried samples), expressing the class membership of the statistical units [22,23,24]. The general architecture of the non-linear model used is a shallow neural network (SNN) [25], which consists of a single hidden layer with a given number of neurons (50) with a rectified linear unit (ReLU) activation function and one output layer with a softmax activation function (normalized exponential function) that normalizes the output in a range of values between 0 and 1. The model was converged using the gradient descent backpropagation algorithm with a learning rate of 0.001, as implemented in the Keras library [26]. The optimal number of hidden neurons was iteratively determined by minimizing the sparse categorical cross entropy, used as a loss function, among the observed and predicted data on the training set, and the accuracy was used as a metric function (performance function) for model performance evaluation. A total of seven datasets were analyzed: five datasets, one for each of the four species (*C. annuum*, 84 samples; *C. baccatum*, 36 samples *C. chacoense*, 12 samples; *C. chinense*, 96 samples), and one inclusive of all species (228 samples) in order to identify the differences between fresh and dried peppers; and two datasets, one for fresh (152 samples) and one for dried (76 samples) peppers. The all-inclusive dataset considers a total of 89 VOCs; each dataset considers a different number of VOCs, including only the ones with a sum different to zero. Each dataset was partitioned into 50% of the samples used for training and cross-validation, while the remaining 50% were used as an internal test. This partitioning was chosen based on the Euclidean distances calculated using the algorithm of Kennard and Stone [27] by selecting parameters without a priori knowledge of a regression model. The best-performing models were selected among the ones with a number of latent vectors (LV) ranging from 1 to 20 [28]. The LV models with the highest mean performance value were the most robust, according to Swierenga et al. [29]. Moreover, for PLSDA, the variable importance in projection (VIP) scores were calculated [30]; for SNN, the variable impact, analogue to VIP, was calculated following the procedure described by Pane et al. [31]. Both indicators were used to estimate the importance of each variable in predicting the correct identity according to each model. The liner models used were developed using a procedure written in the MATLAB 7.1 R14 software. The non-linear models were developed in Python 3.10.12 using Keras API for Tensorflow library, scikit-learn, and OpenCV.

## 3. Results and Discussion

### 3.1. Evaluation of Aroma Compounds from Fresh to Dry Samples

The influence of the drying process on the composition of volatile compounds in one ancestral (*C. chacoense*) and three domesticated pepper species (*C. annuum*, *C. baccatum*, and *C. chinense*) was investigated. Headspace analysis of ripe fruit from different chili pepper cultivars was performed via PTR-ToF-MS on both fresh and dried samples. The VOCs analysis revealed a total of 89 mass spectral peaks obtained from 19 different chili cultivars. The molecules detected belong to the following chemical compound groups: acids, alcohols, aldehydes, esters, furans, hydrocarbons, ketones, terpenes, and sulfur compounds. Each compound was putatively identified on the basis of the high accuracy provided by the tool and by searching the existing literature on the pepper, as already conducted in previous studies [32,33], and the results are reported in Appendix A. Most of the compounds reported here were previously identified as common components in *Capsicum* spp., and some can be used as markers to discriminate different chili species [34,35]. In particular, a maximum of 78 compounds were detected in the fresh samples of “Fatalii”, while a minimum of 33 signals were detected in “Sweet Chocolate”. For the dried samples, a maximum of 56 signals were detected in “Fatalii” and a minimum of 25 in “Jalapeno”. Regardless of the species considered, there was a notable reduction in the complexity of the dried matrix compared to the fresh one, as reported in Table 1. Independently of the matrix considered (fresh or dried), samples belonging to the *C. chinense* species consistently showed the largest number of compounds identified and, generally, the highest signal emission intensities (Table 1). Based on the different aromatic complexity of the fresh samples (understood as the number of different signals identified), we noticed a decrease in the number of total signals emitted in the dry samples, with the exception of the “Sweet Chocolate”. As observed in Table 1, “Chupetinho” showed the most significant difference in the number of signals detected from fresh to dried states (a reduction of 36 signals), followed by “Habanero Red Caribbean” and “Habanero Chocolate” (both with a reduction of 32 signals). Likewise, we observed a strong reduction in the total VOC emission intensity from fresh to dried matrices for all the cultivars considered, except for the two low-capsaicin cultivars “Bell Pepper” and “Sweet Chocolate”, which showed the opposite behavior and the highest VOC emission intensity (Table 1), together with “Habanero Orange”. Indeed, the total amount of emission from the two fresh sweet peppers was in the range of 1700.9 ± 746.7 and 1931.9 ± 584.0 ppbv, respectively (Table 1). The increase in the emission of aroma compounds here observed in the two dried sweet pepper samples was in agreement with the results previously published by Guclu et al. [15]. In Figure 2, both the fresh and dry volatile profiles of two different *Capsicum* species have been reported (*C. annuum*, “Bell Pepper”, and *C. chinense*, “Fatalii”), characterized by distinct aromas, colors, and pungencies. Our results showed the complex effects of drying on volatile compounds emission, where some compounds decrease or disappear while other compounds increase or originate. This emission trend, despite the different patterns, is consistent for all the species and cultivars studied, confirming the previous observations of Toontom et al. [13] in *C. anuum*.

Indeed, it is known that during the dehydration process, several aromatic compounds are subjected to changes due to sequential reactions, including enzymatic hydrolysis, thermal degradation, unsaturated fatty acid degradation, and Maillard reactions, all of which can alter the volatile profile of the dried product [11]. These reactions seem to make the volatile profile of dried samples less complex compared to the fresh ones by causing the complete loss of, or a strong reduction in, the emission of some compounds. For example, the application of drying processes resulted in the loss of signals detected at *m*/*z* 30.038 (tentative identification ethylene) *m*/*z* and 47.049 (TI ethanol), which are both low-molecular-weight compounds associated with the ripening process of the fruit on the plant [36]. On the contrary, we observed a decrease in other low-molecular-weight signals detected at *m*/*z* 31.018 (TI Formaldehyde), *m*/*z* 33.033 (TI methanol), *m*/*z* 45.033 (TI acetaldehyde), and *m*/*z* 47.013 (formic acid/formates), which are linked to compounds that occur naturally in some fruits as a product of metabolism following the drying process (Figure 2 and Appendix A). A similar behavior was observed for terpene and terpenoid compounds (detected at *m*/*z* 137.132, 153.126, 155.145, and 205.195, respectively), which are linked to the fruity notes. Moreover, terpene compounds showed a great diversity both among species and cultivars of the same species, with higher emissions in the cultivars of *C. chinense* compared to the other *Capsicum* samples in both fresh and dried matrices. Likewise, the emission intensity of the aforementioned compound fragments (detected at 67.054, 81.069, 83.086, 93.069, 107.086, 109.101, 121.101, and 123.117) decreased by over 90% in the dried samples compared to the fresh ones. As reported by Guclu et al. [15], the reduced emission observed in dry samples could be due to the high temperature applied during the drying process, which can result in the degradation of many aromatic compounds. On the contrary, acetic acid (*m*/*z* 61.028), which was the most abundant acid in both the fresh and dry samples, showed an increase in emission following the drying process. This behavior was common across all pepper species studied except for *C. chinense,* where the emission of acetic acid was considerably higher in the fresh matrix. Similarly, the increased emission of 2-methylpropanal (*m*/*z* 73.065) and acetone (*m*/*z* 59.049) could be due, respectively, to the heat breakdown of methionine by the Strecker reaction [37] or linked to degradation products of sugars and carotenoids [38]. As reported by different authors [39,40,41], the decomposition of sulfur-containing amino acids via the Maillard reaction induces an increased emission of sulfur compounds in dried samples, as highlighted by the high signals detected at *m*/*z* 46.997, *m*/*z* 48.003, *m*/*z* 49.011, *m*/*z* 61.011, and *m*/*z* 63.027. These compounds, released by the action β-lyases enzymes on the cysteine-S-conjugates, typically have a low detection threshold in humans and are recognized as important aroma contributors in dried bell peppers [42]. It is interesting to note that the emission amount of sulfur compounds (both in terms of total emission and the number of different signals) differed among *Capsicum* species and cultivars (Figure 2, Appendix A), and this could help to develop powders with different aromas and tastes. For example, methanethiol showed the lowest emission intensity in all *C. baccatum* cultivars (averaging under 10 ppbv), while the other *Capsicum* species showed a higher emission intensity (above 10 ppbv). In contrast, dimethyl sulfide, *m*/*z* 63.027, showed a similar emission intensity in all cultivars (about 30 ppbv) except for “Bell pepper”, “Jalapeno”, “Habanero Yellow”, and “Monkey’s Nipple”, which showed significantly higher content, while “Chupetinho” and “Hot Lemon” exhibited a lower content. “Bell Pepper” and “Jalapeno”, as well as “Monkey’s Nipple” and “Habanero chocolate”, were the cultivars with the highest methanethiol and thioacetaldehyde emission.

Moreover, our analysis highlighted how the species/varieties of *Capsicum* do not always show the same behavior. For example, in the two sweet peppers (“Bell pepper” and “Sweet Chocolate”), some volatile compounds derived from carotenoids degradation, lipid oxidation, or the Stecker reaction were higher in dried samples compared to fresh ones (Table 1, Figure 2). This result, mostly observed in the two sweet peppers studied, is in agreement with Jun and co-authors [43], where some compounds (e.g., acetaldehyde, acetone, 3-methyl butanal, etc.) showed higher levels in dried samples. Thus, the volatile profile from fresh to dried samples changes drastically under heating, modifying the aroma of dried pepper powder, regardless sweet or spicy. However, the aromatic notes differed widely among *Capsicum* species, mainly in fresh rather than dried matrices. To further evaluate differences and similarities and to highlight the drying process effects on the aroma composition of different species, a PLSDA was applied on the detected VOC. As expected, from fresh to dried samples, we observed a deep change in aroma, as confirmed by the PLSDA result. In particular, we learned how the drying process differently affected the composition of volatile compounds among different pepper species and cultivars. Table 2 shows the results of the VOCs-based PLSDA model for classifying fresh and dry samples within each species and for the whole dataset. In detail, for the single species, the models demonstrated excellent performances in terms of the percentage of correct classification (100%) and low errors (classification error and RMSEC). The entire model, which considers all the samples, also showed very high performance (98.2%). The performance, in terms of the percentage of correct classification in the independent test set, was also excellent (100%) for the single species and slightly lower for the total model (94.7%).

In Figure 3, the plot of the PLSDA scores of all 228 pepper samples is shown, grouped according to their status (dried or fresh) and representative of the first two LVs (as detailed in Table 2). Considering that the whole model is composed of six LVs, the first two (x-block 53.06%; y-block 76.41% of cumulated variance) could still return a partial separation among the groups, for this reason few fresh samples appear to overlap with the dried ones (mainly Chacoence and Sweet Chocolate).

Similar results have been shown for non-linear SNN models (Table 3) with excellent performances in terms of the percentage of correct classification (always 100%) in the training set. However, there were slightly lower performances for *C. annuum* (90.5%) and the total dataset (98.2%) in the test sets.

Thus, VIP scores (higher than 1) were used to estimate the importance of each variable for each species in the PLS-DA model. The VIP scores obtained by the PLSDA models used to classify fresh and dried pepper samples were different among *Capsicum* species (Table 4). Table 4 lists the compounds with higher VIP scores, which vary the most between the fresh and dry matrices for each species considered.

In particular, excluding fragment compounds (*m*/*z* 27.22, 41.038, 43.054, and 57.033), the VIP scores obtained by the PLSDA models used to differentiate between fresh and dried pepper samples were *m*/*z* 30.038 (TI as ethylene), 45.033 (TI acetaldehyde), 46.994 (TI sulfurate fragments/thioformaldehyde), 47.013 (TI formic acid/formates), 47.049 (TI ethanol), 48.003 (TI methanethiol), 61.011 (TI thioacetaldehyde), 63.027 (TI dimethyl sulfide), 73.065 (TI 2-methylpropanal), 97.064 (TI 2-ethylfuran), 103.075 (TI 3-methylbutanoic acid), 115.111 (TI heptanal), and 205.195 (TI sesquiterpenes) (Appendix A). These VIP scores highlighted how the powder obtained by different *Capsicum* species was characterized by odor compounds with different impacts and thresholds and therefore a distinct final flavor profile. Indeed, the heating process led, for example, to the generation of thermally derived aroma compounds with low odor thresholds, such as many S compounds (*m*/*z* 46.994, 48.003, 61.011, and 63.027), which were detected in all dried samples but not in the fresh ones. Despite the relatively low amounts of the S-compounds identified in the *Capsicum* samples, their presence has great importance in forming the characteristic scent of their powder, as suggested by Korkmaz et al. [5]. On the contrary, high temperature leads some compounds to either disappear, as, for example, did the signal detected at *m*/*z* 30.038 (TI ethylene) and 47.049 (TI ethanol), which was emitted by all fresh samples and not from dried ones, or to deeply decreased, as observed in the signal detected at 97.064, 103.075, 115.111, and 205.195 (TI sesquiterpenes).

Finally, acetaldehyde, which was the main aldehyde detected in all samples, showed an emission that was at least ten times higher in *C. chacoense* (wild species) compared to the other species. It is well known that acetaldehyde can contribute to the sweetness or fruity odor depending on its emission intensity. Indeed, as reported by Liu et al. [44], acetaldehyde provides fruity attributes at lower concentrations, while tending to suppress positive odors when occurring at higher concentrations. This could suggest that during the process of domestication, there may have been a selective pressure favoring chili pepper varieties with lower concentrations of acetaldehyde, thus enhancing other desirable aromas for human consumption.

### 3.2. Dried Chili: Volatile Markers for Species Discrimination

In this section, we discuss the volatile organic compounds identified through a partial least squares discriminant analysis (PLSDA) approach, which could serve as volatile markers for discriminating between dried chili pepper species. The same authors previously successfully applied this method to identify volatile markers for each species in the fresh pepper matrix [34]. In detail, both models (fresh and dried) exhibited excellent performance (Table 5), with a 100%-correct classification rate and low errors (classification error and RMSEC). When considering the independent test sets, the performance in terms of correct classification percentage was very high for the fresh samples (98.6%), while for the dried ones, it was lower (89.5%).

In Figure 4, the plot of the PLSDA scores of the 76 dried pepper samples is reported (19 cultivars per 4 replicates) and grouped according to their species identity, representative of the first two LVs. Considering that the whole model is composed of nine LVs, the first two (x-block 29.69%; y-block 40.23% of cumulated variance) still return a partial separation among the groups. 

The aroma of dried chili peppers can vary due to factors such as species, variety, drying method, and storage conditions. Consequently, several aromatic compounds are released during drying, while others may be altered or lost [5,15]. Thus, owing to their specific aromatic composition, the dried peppers from each *Capsicum* species showed a separate dispersion in the graphical space (Figure 4). However, while almost all the samples belonging to the three domesticated species clearly separate from each other, the wild species, interestingly, is positioned at the center and partially overlaps with other species in the graph (Figure 3). Thus, our results suggest that different pepper accessions contain unique combinations and concentrations of volatile compounds, contributing to their distinct aromas. Therefore, even when dried, different chili species retain some of their key compound aromas that differ among species and varieties. Table 6 lists VOCs with higher variable importance in projection (VIP) values (>1.5), which could be promising candidates for dried pepper classification or highlighting common aromatic traits within species. These key chemical species for species classification were detected at (1) *m*/*z* 89.059, 109.101, and 205.195 for *C. annuum*; (2) *m*/*z* 61.011, 95.049, and 117.094 for *C. baccatum*; (3) *m*/*z* 87.045, 95.049, and 121.014 for *C. chacoense*; and (4) *m*/*z* 109.101, 119.085, and 205.195 for *C. chinense*. Among these VIPs (Table 6), the signal detected at *m*/*z* 89.059 (tentatively identified as ethyl acetate) emerged as a key compound for *C. annuum* accessions, despite its low emission intensity. Ethyl acetate is a sweet-smelling organic compound with a fruity aroma, which may interact with other aroma compounds present in the peppers, influencing their overall aroma profile. Conversely, the *m*/*z* detected at 109.101, 119.085, and 205.195 are signals associated with aromatic terpene-class compounds [45,46]. These compounds, which contribute to the aroma and flavor of peppers, were often detected in many pepper accessions, both in fresh and dried matrices [5,34]. However, the emission intensity of these signals decreases in the dried matrix, varying depending on the pepper species and variety. As reported in Table 6, terpene compounds (*m*/*z* 205.195) and their fragments (*m*/*z* 109.101, 119.085) represent key aromas for the dried powder from both *C. chinense* and *C. annuum* species. Specifically, these signals exhibited very low emission intensity in each *C. annuum* cultivar, while consistently appearing at high levels in *C. chinense* cultivars. This difference resulted in both chemical species being identified as important VIP factors for their classification. Additionally, the *m*/*z* 205.195 signal was also identified as an important VIP for *C. chacoense*. Additionally, the signal detected at *m*/*z* 87.045 (TI as 2,3-butanedione/diacetyl) has emerged as a volatile marker for the wild accession, alongside *m*/*z* 121.034 (TI as 3-methyl-5-propyl-1,2-dithiolane) which, depending on the concentration, can impart bell pepper-like fruity notes (low concentration) or sulfurous, onion, and mushroom-like aromas (high concentration) [47]. The 2,3-butanedione, a volatile organic compound renowned for its buttery or creamy flavors, was detected in trace amounts in dried chili peppers. However, its emission was notably higher in the wild accession compared to domesticated varieties, suggesting its potential contribution to subtle buttery notes within the overall aroma profile. Furthermore, S compounds (*m*/*z* 61.011 and 94.998) emerged as key compounds for the classification of *C. baccatum* peppers, alongside hexanoic acid (*m*/*z* 117.094), as detailed in Table 6. Although the emission of S compounds was lower in *C. baccatum* peppers, hexanal emissions were significantly higher compared to other *Capsicum* species. Hexanoic acid, a naturally occurring fatty acid found in various plants including the *Capsicum* species [48], plays a role in the plant’s defense mechanisms against pathogens. Additionally, it contributes to the overall aroma profile of dried *C. baccatum* peppers, imparting nuances of freshness when dried. Comparing the findings of this study with prior research [34], certain VIPs, particularly 2,3-butanedione (*m*/*z* 87.045), hexanoic acid/hexanoates (*m*/*z* 117.094), and sesquiterpenes (*m*/*z* 205.195), emerged as important for characterizing both fresh and dry samples.

## 4. Conclusions and Future Perspectives

This research has focused on analyzing the volatile compound profiles of different species and cultivars of chili peppers, both fresh and dried. The data obtained revealed significant variations in the composition and concentration of VOCs in both fresh ripened fruit and dried powder obtained from various *Capsicum* species (*annuum*, *baccatum*, *chacoense*, and *chinense*), with each species exhibiting unique dispersion patterns in graphical representations. As expected, there was a noticeable aroma change from fresh to dried samples. Fresh peppers exhibited higher VOC content compared to dried ones, except for the two sweet peppers studied, which showed the opposite behavior. Chili peppers belonging to the species *C. chinense* typically exhibited a higher number of compounds and a higher total signal emission intensity compared to other species. While the domesticated species show clear separation from each other based on their volatile compound profiles, the wild species *C. chacoense* exhibits a partial overlap with other species, indicating its distinct characteristics; for example, it was characterized by a much higher emission of acetaldehyde compared to all the other samples. Through the PLSDA approach, several findings emerged: (1) that the effect of drying on volatile compound composition varies among different pepper species and sometimes even among different cultivars; (2) key compounds that can be used for species classification in dried powder. 

This research serves as a preliminary guideline for promoting the utilization of different pepper species and cultivars to enhance pepper powder taste and product diversity and valorization. 

In conclusion, this study underscores the significant aromatic potential inherent in each variety of chili pepper, with each possessing its own distinct aroma and flavor profile. This diversity allows for the creation of various dried powder products suitable for a wide range of applications. Whether seeking to add heat, depth, or nuanced flavor in dishes, the selection of chili pepper options provides ample opportunities for culinary creativity and exploration. This work lays down the foundation for future studies to have a better understanding of the aromatic volatile profile of chili peppers and could help to focus on which specific factors primarily affect the flavor of pepper species and cultivars, such as geographical location or growing conditions. Additionally, it would be interesting to investigate the effects of different storage methods and durations on preserving the optimal aroma. Lastly, it would be appropriate to incorporate sensory analyses into these studies, both through expert panels and consumer-focused evaluations, to highlight preferences and explore potential innovative and trendy culinary uses. This study aims to promote biodiversity, conservation and the promotion and valorization of typical varieties with unique flavors.

## Figures and Tables

**Figure 1 plants-13-01131-f001:**
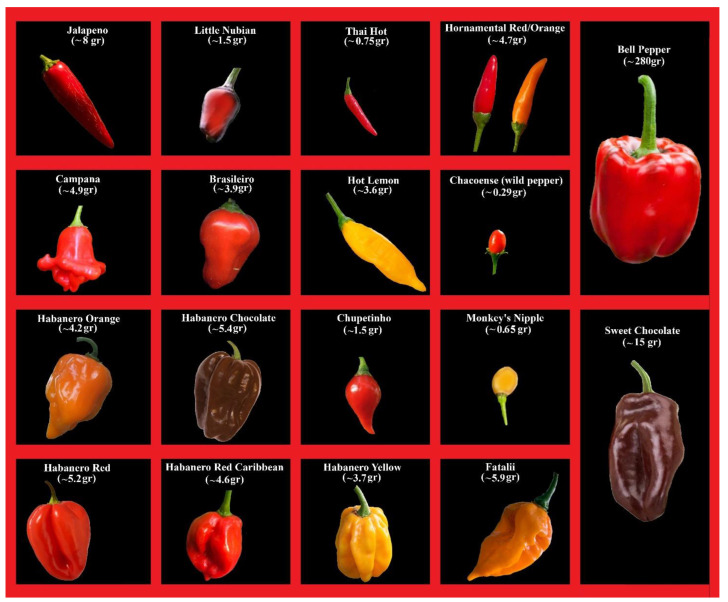
Morphological biodiversity of the chili cultivars studied. The selected *Capsicum* germplasm showed different phenotypes for fruit: morphology, color, weight, and pungency [4]. The cultivars in the first row are *C. annuum* genotypes, the second row contains *C. baccatum* and *chacoense*, and the third and fourth rows contain *C. chinense* accessions. Within the larger cells, the genotypes with the lowest capsaicin levels are identified.

**Figure 2 plants-13-01131-f002:**
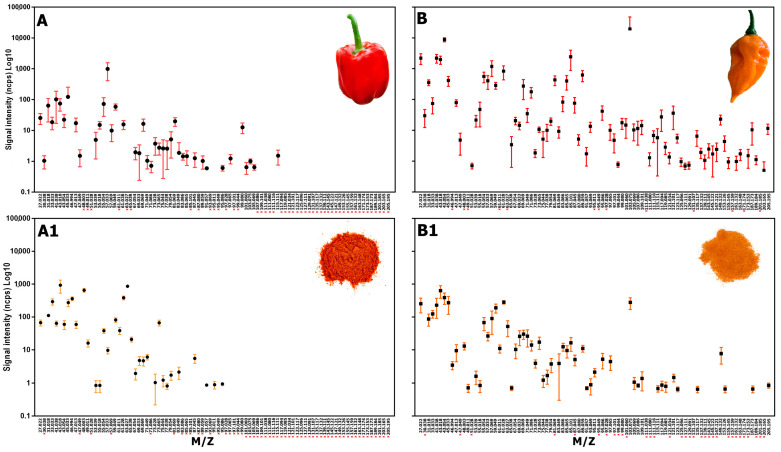
Example PTR-TOFMS mass spectrum detected on fresh and dry sample for (**A**,**A1**) *C. annuum* (“Bell Pepper”) and (**B**,**B1**) *C. chinense* (“Fatalii”). Absent signals are highlighted with a red “x” at the bottom of each box.

**Figure 3 plants-13-01131-f003:**
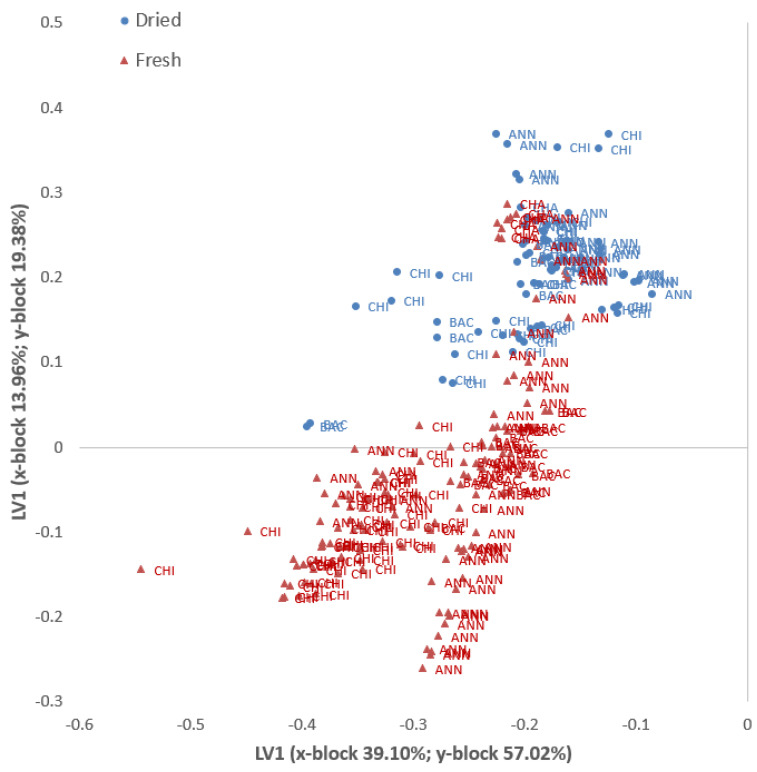
Representation of fresh and dried pepper samples on the first two axes of the PLSDA model composed of nine LVs.

**Figure 4 plants-13-01131-f004:**
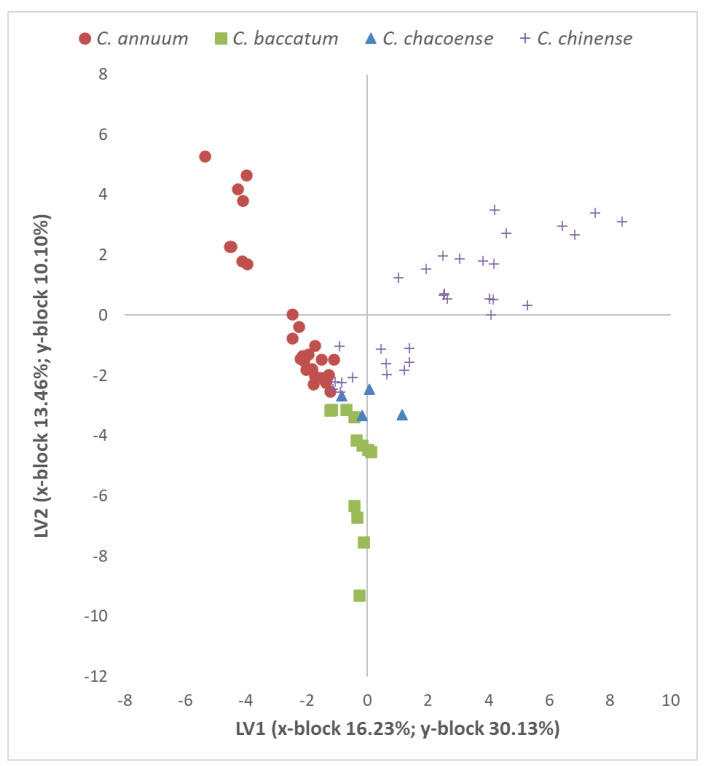
Representation of dried pepper samples on the first two axes of the PLSDA model composed of nine LVs.

**Table 1 plants-13-01131-t001:** Total VOC emissions, total signals detected, and the difference in the number of signals detected from fresh to dry matrices recorded for each pepper sample. Data have been grouped according to the *Capsicum* species studied.

Species	Varieties	*Capsicum* Fresh Sample			*Capsicum* Dry Sample			From Fresh to Dry	
		Total VOC Emissions (Average and SD)	Total VOCs Emission of Each Species (Average)	Total Signals Detected (N°)	Total VOC Emissions (Average and SD)	Total VOC Emissions of Each Species (Average)	Total Signals Detected (N°)	Total VOCs Emissions Reduction/Increase (%)	Difference of Number of Signals Detected from Fresh to Dry
*C. chacoense*	Chacoense	18,572.5 ± 2410.0	18,572.5	52	1333.3 ± 194.3	1333.3	32	−92.8	20
*C. annuum*	Bell Pepper	1700.9 ± 746.7	5414.7	40	4424.4 ± 386.2	1949.1	34	160.1	6
	Little Nubian	10,783.5 ± 3231.7		60	2247.3 ± 676.1		31	−79.2	29
	Jalapeno	11,905.1 ± 1674.7		50	954.2 ± 133.8		25	−92.0	25
	Orange Hornamental	4117.8 ± 748.8		45	1515.9 ± 358.8		36	−63.2	9
	Red Hornamental	2588.9 ± 616.1		46	629.9 ± 71.9		33	−75.7	13
	Sweet Chocolate	1931.9 ± 584.0		33	3979.5 ± 355.4		41	106.0	−8
	Thai Hot	4875.4 ± 827.0		46	1111.8 ± 141.4		29	−77.2	17
*C. baccatum*	Brasileiro	6507.8 ± 1418.0	11,191.0	62	929.5 ± 165.9	844.9	43	−85.7	19
	Campana	10,137.5 ± 1090.6		55	1027.9 ± 284.5		26	−89.9	29
	Hot lemon	16,303.5 ± 2880.5		59	577.4 ± 84.0		30	−96.5	29
*C. chinense*	Monkey’s Nipple	9012.6 ± 750.6	23,693.0	60	718.4 ± 112.9	2353.1	35	−95.6	25
	Chupetinho	36,121.7 ± 3833.3		67	1140.7 ± 113.0		31	−96.8	36
	Fatalii	34,114.1 ± 3131.6		78	3232.9 ± 440.9		56	−90.5	22
	H. Chocolate	36,472.8 ± 5335.1		78	1917.5 ± 331.4		46	−94.7	32
	H. Orange	29,084.2 ± 5042.9		72	5148.1 ± 584.4		43	−82.3	29
	H. Red	18,841.1 ± 2247.7		65	2801.5 ± 353.3		50	−85.1	15
	H. Red Caribbean	29,179.2 ± 1146.2		73	1383.7 ± 196.3		41	−95.3	32
	H. Yellow	23,646.0 ± 4056.2		62	2482.2 ± 250.2		49	−89.5	13

**Table 2 plants-13-01131-t002:** Results of the PLSDA models based on VOCs used to classify fresh and dried pepper samples.

	*C. annuum*	*C. baccatum*	*C. chacoense*	*C. chinense*	Total
N	84	36	12	96	228
N. units (X-block)	70	72	56	89	89
N. units (Y-block)	2	2	2	2	2
Preprocessing	Autoscale	Autoscale	Autoscale	Autoscale	Normalize
N. LV	2	2	2	3	6
% Cumulated variance X-block	50.13	62.64	72.10	52.79	82.01
% Cumulated variance Y-block	37.92	42.61	44.40	41.56	87.74
Mean specificity	1	1	1	1	0.97
Mean sensitivity	1	1	1	1	0.99
Random probability (%)	50	50	50	50	50
Mean class. err.	0	0	0	0	0.02
Mean RMSEC	0.69	0.51	0.50	0.68	0.25
%Corr. class. model	100	100	100	100	98.2
% Corr. class. independent test	100	100	100	100	94.7

N = number; LV = latent vectors; RMSEC = root mean square error of calibration.

**Table 3 plants-13-01131-t003:** Results of the SNN models based on VOCs used to classify fresh and dried pepper samples.

	*C. annuum*	*C. baccatum*	*C. chacoense*	*C. chinense*	Total
N	84	36	12	96	228
N. units (X-block)	70	72	56	89	89
N. units (Y-block)	2	2	2	2	2
Random probability (%)	50	50	50	50	50
% Corr. class. model	100	100	100	100	100
% Corr. class. independent test	90.5	100	100	100	98.2

N = number.

**Table 4 plants-13-01131-t004:** Top five VOCs with higher VIP scores obtained by the PLSDA models used to classify fresh and dried pepper samples.

	*C. annuum*	*C. baccatum*	*C. chacoense*	*C. chinense*	Total
#1	61.011 (TI Thioacetaldehyde)	30.038 (TI Ethylene isotope)	63.027 (TI Dimethylsulfide)	61.011 (TI Thioacetaldehyde)	43.054 (TI Alkyl fragment (alcohols))
#2	30.038 (TI Ethylene isotope)	47.049 (TI Ethanol)	46.994 (TI Thioformaldehyde)	46.994 (TI Thioformaldehyde)	103.075 (TI Ethyl 3-methylbutanoate/3-methylbutanoic acid)
#3	73.065 (TI Butanone/butanal)	63.027 (TI Dimethylsulfide)	48.003 (TI Methanethiol)	30.038 (TI Ethylene isotope)	57.033 (TI C3 Aldehyde and ketone fragments)
#4	115.111 (TI Heptanal)	48.003 (TI Methanethiol)	47.013 (TI Formic Acid/Formates)	103.075 (TI Ethyl 3-methylbutanoate/3-methylbutanoic acid)	41.038 (TI Alkyl fragment (alcohols and esters)
#5	47.049 (TI Ethanol)	27.022 (TI Alkyl fragment)	205.195 (TI Sesquiterpenes)	97.064 (TI 2-Ethylfuran)	45.033 (TI Acetaldehyde)

**Table 5 plants-13-01131-t005:** Results of the PLSDA models based on VOCs used to classify species within fresh and dried pepper samples.

	Fresh	Dried
N	152	76
N. units (X-block)	84	64
N. units (Y-block)	4	4
Preprocessing	Autoscale	Autoscale
N. LV	10	9
% Cumulated variance X-block	76.94	76.58
% Cumulated variance Y-block	60.92	60.74
Mean specificity	0.99	0.99
Mean sensitivity	1	1
Random probability (%)	25	25
Mean class. err.	0.002	0.004
Mean RMSEC	0.39	0.38
% Corr. class. model	100	100
% Corr. class. independent test	98.6	89.5

N = number; LV = latent vectors; RMSEC = root mean square error of calibration.

**Table 6 plants-13-01131-t006:** List of the key compounds for species classification of dry samples obtained by the PLSDA models. VIP scores higher than 1.5 have been highlighted with an asterisk.

N° Compounds	*m*/*z*	Chemical Formulae	Tentatively Identification	VIP Scores			
				*C. annuum*	*C. baccatum*	*C. chacoense*	*C. chinense*
1	61.011	C_2_H_5_S^+^	S compound (Thioacetaldehyde)	1.233	* 1.579	1.049	0.957
2	87.045	C_4_H_7_O^+^	2,3-Butanedione/Diacetyl	0.534	0.951	* 2.400	0.655
3	89.059	C_4_H_9_O^+^	Ethyl acetate	* 1.510	0.478	0.205	1.510
4	94.998	C_2_H_7_OS_2_^+^	S compound Dimethyl disulfide	0.836	* 1.559	0.749	0.758
5	109.101	C_8_H_13_^+^	Terpenes fragments	* 1.749	1.077	0.760	* 1.862
6	117.094	C_6_H_13_O_2_^+^	Hexanoic acid/hexanoates	0.827	* 1.721	1.213	0.850
7	119.085	C_9_H_11_^+^	Terpenes fragments	1.484	1.028	0.898	* 1.581
8	121.014	C_4_H_9_S_2_^+^	3-methyl-5-propyl-1,2-dithiolane	0.923	1.218	* 2.807	0.746
9	205.195	C_15_H_25_^+^	Sesquiterpenes	* 1.695	0.494	* 1.319	* 1.772

## Data Availability

Data are contained within the article and Appendix A.

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
