# Peer review of "Influence of the Drying Process on the Volatile Profile of Different Capsicum Species"

_plants, 2024, doi:10.3390/plants13081131_

Round 1
Reviewer 1 Report
Comments and Suggestions for Authors
The article shows differences in the content of volatile compounds analyzed by PRT-TOF-MS, between the fresh and dry state of different Capsicum species. Although the theme may not be something innovative, it uses different species of chilli using a technique that is not very common for the analysis of this type of compounds (volatiles) and using statistical classification analysis (PLS-DA) to discriminate the compounds that allow us to classify between the dry and fresh chili samples and to discriminate among the varieties studied.
Despite what has been mentioned, there are some issues that I would like to see clarified:
Keywords: very repetitive keywords: eliminate dried capsicum and aromatic composition. Include PLSDA
Review the text, there are minimal English errors (Examples: Figure 1, the abbreviations of the numbers are not correct, be careful with between and among, etc.).
Material and methods:
Lines 86-97. Include more information about sample collection (time of year for each species, location, etc.)
Results and discussion:
Table 1. Apply statistics to the data obtained in this table.
Tables 2,3, 5. Include the name of the acronyms at the bottom of the table (e.g. N, n., n.LV, RMSEC).
Lines 306-340. Include the name of the compounds tentatively identified as markers through their VIP scores in Table 4 (not just in the text). Clearly place in the table the compounds that can discriminate fresh samples from dried samples. Furthermore, to choose these compounds as markers, have the authors only taken into account the VIP value? Have they not taken into account the significant differences found between fresh and dry samples? The same question arises for the classification carried out to differentiate between chilli species (Table 6).
In addition to table 4, I would like you to include a figure like the one shown in Figure 3, to be able to see the different classification of samples depending on whether the samples are in a fresh or dry state.
Uniformity in the text in the placement of masses with decimals.
Rewrite the conclusions, they sound too generic and do not take into account the specific results obtained in this work.
Comments on the Quality of English Language
The article shows differences in the content of volatile compounds analyzed by PRT-TOF-MS, between the fresh and dry state of different Capsicum species. Although the theme may not be something innovative, it uses different species of chilli using a technique that is not very common for the analysis of this type of compounds (volatiles) and using statistical classification analysis (PLS-DA) to discriminate the compounds that allow us to classify between the dry and fresh chili samples and to discriminate among the varieties studied.
Despite what has been mentioned, there are some issues that I would like to see clarified:
Keywords: very repetitive keywords: eliminate dried capsicum and aromatic composition. Include PLSDA
Review the text, there are minimal English errors (Examples: Figure 1, the abbreviations of the numbers are not correct, be careful with between and among, etc.).
Material and methods:
Lines 86-97. Include more information about sample collection (time of year for each species, location, etc.)
Results and discussion:
Table 1. Apply statistics to the data obtained in this table.
Tables 2,3, 5. Include the name of the acronyms at the bottom of the table (e.g. N, n., n.LV, RMSEC).
Lines 306-340. Include the name of the compounds tentatively identified as markers through their VIP scores in Table 4 (not just in the text). Clearly place in the table the compounds that can discriminate fresh samples from dried samples. Furthermore, to choose these compounds as markers, have the authors only taken into account the VIP value? Have they not taken into account the significant differences found between fresh and dry samples? The same question arises for the classification carried out to differentiate between chilli species (Table 6).
In addition to table 4, I would like you to include a figure like the one shown in Figure 3, to be able to see the different classification of samples depending on whether the samples are in a fresh or dry state.
Uniformity in the text in the placement of masses with decimals.
Rewrite the conclusions, they sound too generic and do not take into account the specific results obtained in this work.
Reviewer 2 Report
Comments and Suggestions for Authors
The manuscript investigates the impact of the drying process on the aromatic composition of various Capsicum species, including both wild and domesticated peppers. Using a headspace PTR-TOF-MS, the aroma of 19 different pepper samples belonging to four species was analyzed. Results indicate significant changes in the composition and concentration of volatile organic compounds (VOCs) from fresh to dried capsicum. In my opinion, authors should address the following issues before considering this manuscript for publication:
· The authors should discuss the limitations of PTR-ToF-MS in terms of its capability to detect particular VOCs. Given that only compounds with proton affinity higher than water can be protonated and analyzed, what are the implications of this limitation, and what complementary techniques could be employed to overcome it?
· It is essential to highlight the main compounds that discriminate dry Capsicum samples and compare them with the previous study where the authors did the same for fresh samples. This comparison would provide a more comprehensive understanding of the effects of the drying process on aromatic composition.
· The authors should justify the use of such a large number of latent variables (9 LVs) for the PLS-DA model. Providing a rationale for this choice would enhance the transparency and reproducibility of the analysis.
· Why dataset was split 50/50? Usually training set is larger than the test set
· PLS-DA classifies samples based on their origin. Based on Table 4 title, one can conclude that this model serves to discriminate dry from fresh samples. It is clear that we don’t need chemical analysis to do this.
· In Table 6, delete tag “S compound”. It is clear that Thioacetaldehyde contains S atom. Moreover, suggest the identity of the compound at m/z 121.034.
· The information provided in Figure 2 is currently not useful due to small graphs and labels. The authors should enlarge the graphs and labels to improve readability and comprehension.
Comments on the Quality of English LanguageManuscript requires language editing. For example, line 38: “includes” should be replaced with “, which includes” ; Line 44: delete “chemical compounds” ; Line 46: “average consumption recommended” should be replaced with “average recommended consumption” ; Line 85 – check this subheading title; Line 121-122 – delete one “samples” ; Line 126-127 – not a clear sentence; Line 137- not a clear sentence; Line 142 – “statistic” should be replaced with “statistical” ; Line 214 – Edit this sentence; Check Table 21 caption, one “.” should be removed. Format Table 1, change Table orientation to make it more clear; Line 255 - delete “respectively”; Line 259 – “change sulfur containing to sulfur-containing” and simplify the sentence. Line 334: “contributed” should be replaced with “ contribute”; Line 337 - delete “ could”
Round 2
Reviewer 1 Report
Comments and Suggestions for Authors
The authors have responded adequately to the questions raised, therefore I consider that the manuscript would be ready for acceptance.
Reviewer 2 Report
Comments and Suggestions for Authors
The authors have addressed all my comments. The revised version of the manuscript is now suitable for publishing in Plants.